# A Novel Method for Angiographic Contrast-Based Diagnosis of Stenosis in Coronary Artery Disease: In Vivo and In Vitro Analyses

**DOI:** 10.3390/diagnostics14131429

**Published:** 2024-07-04

**Authors:** Woongbin Kang, Cheong-Ah Lee, Gwansuk Kang, Dong-Guk Paeng, Joonhyouk Choi

**Affiliations:** 1Faculty of Earth and Marine Convergence, Major Ocean Systems, Jeju National University, Jeju 63243, Republic of Korea; ubin1021@jejunu.ac.kr (W.K.); a2014106251@jejunu.ac.kr (C.-A.L.); 2Division of Gastroenterology and Hepatology, Stanford University School of Medicine, Stanford, CA 94305, USA; gwansuk@stanford.edu; 3Focused Ultrasound Foundation, Charlottesville, VA 22903, USA; 4Division of Cardiology, Department of Internal Medicine, School of Medicine, Jeju National University, Jeju 63241, Republic of Korea

**Keywords:** coronary artery disease, coronary angiography, stenosis, angiography, contrast media

## Abstract

Background: The existing diagnostic methods for coronary artery disease (CAD), such as coronary angiography and fractional flow reserve (FFR), have limitations regarding their invasiveness, cost, and discomfort. We explored a novel diagnostic approach, coronary contrast intensity analysis (CCIA), and conducted a comparative analysis between it and FFR. Methods: We used an in vitro coronary-circulation-mimicking system with nine stenosis models representing various stenosis lengths (6, 18, and 30 mm) and degrees (30%, 50%, and 70%). The angiographic brightness values were analyzed for CCIA. The in vivo experiments included 15 patients with a normal sinus rhythm. Coronary angiography was performed, and arterial movement was tracked, enabling CCIA derivation. The CCIA values were compared with the FFR (*n* = 15) and instantaneous wave-free ratio (iFR; *n* = 11) measurements. Results: In vitro FFR showed a consistent trend related to the length and severity of stenosis. The CCIA was related to stenosis but had a weaker correlation with length, except for with 70% stenosis (6 mm: 0.82 ± 0.007, 0.68 ± 0.007, 0.61 ± 0.004; 18 mm: 0.78 ± 0.052, 0.69 ± 0.025, 0.44 ± 0.016; 30 mm: 0.80 ± 0.018, 0.64 ± 0.006, 0.40 ± 0.026 at 30%, 50%, and 70%, respectively). In vitro CCIA and FFR were significantly correlated (R = 0.9442, *p* < 0.01). The in vivo analysis revealed significant correlations between CCIA and FFR (R = 0.5775, *p* < 0.05) and the iFR (*n* = 11, R = 0.7578, *p* < 0.01). Conclusions: CCIA is a promising alternative for diagnosing stenosis in patients with CAD. The initial in vitro validation and in vivo confirmation in patients demonstrate the feasibility of applying CCIA during coronary angiography. Further clinical studies are warranted to fully evaluate the diagnostic accuracy and potential impact of CCIA on CAD management.

## 1. Introduction

Coronary artery disease (CAD) is a global health concern with high morbidity and mortality rates and is characterized by atherosclerotic stenosis and occlusion of the coronary arteries, resulting in reduced blood flow to the myocardium and potential complications such as angina pectoris and heart failure. The current diagnostic methods for stenosis, namely coronary angiography, using contrast media, and fractional flow reserve (FFR), using specialized wires and drugs, have limitations, including their invasiveness, high cost, and time consumption, resulting in discomfort and other adverse effects. Recently, a virtual fractional flow reserve (vFFR) method using Navier–Stokes equations was developed [1,2]. However, these equations are not equivalent to the FFR, as they cannot fully represent the coronary microvascular circulation (CMVC) [3].

To address these limitations, we propose an innovative and less invasive approach to diagnosing stenosis. Our novel method, i.e., coronary contrast intensity analysis (CCIA), introduces a new measurement approach and physiological factors. CCIA calculates the blood flow ratio by analyzing the changes in the contrast agent concentration over time in the proximal and stenotic regions. In this study, we aimed to assess the feasibility and accuracy of CCIA in diagnosing stenosis and compare CCIA with FFR. While FFR uses pressure to measure the blood flow, CCIA is derived from observing the blood flow and is expected to reflect the CMVC.

Our primary objective in conducting these in vitro experiments was to introduce a diagnostic method that is more efficient, less invasive, and rooted in a robust theoretical foundation. This foundational step is crucial to ensure the reliability and validity of the proposed method before its translation into clinical practice. We believe our study will introduce a more efficient and less invasive diagnostic method using blood flow for stenosis in patients with coronary artery diseases and ultimately enhance patient comfort and reduce healthcare costs.

## 2. Materials and Methods

### 2.1. The In Vitro Experiment Design

To examine the viability of this innovative approach and its associated physiological metrics, we conducted experiments involving CCIA and FFR measurements. To assess the effectiveness of our proposed diagnostic method, we designed and implemented an in vitro experiment to replicate the intricacies of human coronary circulation. The decision to conduct in vitro experiments prior to transitioning the method into clinical practice was based on the absence of analogous research within the scope of our experimental design. Measurements were performed using various stenosis models and in vitro circulatory systems. Using this experimental setup, we calculated the CCIA value for each case and compared the results with those of FFR, which is the gold standard. Details of the experimental setup are discussed below.

### 2.2. Theoretical Background

FFR is defined as the ratio between the maximum blood flow in the stenotic artery and the normal maximum blood flow. The calculation of the FFR involves pressure measurements, as depicted in a previous study [4].
$$FFR=\frac{Q_{S\;max}}{Q_{N\;max}}=\frac{\frac{\left(P_{d}-P_{v}\right)}{R_{S\;max}}}{\frac{\left(P_{a}-P_{v}\right)}{R_{N\;max}}}=\frac{\left(P_{d}-P_{v}\right)}{\left(P_{a}-P_{v}\right)}=\frac{P_{d}}{P_{a}}\;$$

(*P_a_*, aortic pressure; *P_d_*, distal coronary pressure; *P_v_*, venous pressure; *Q_S max_*, hyperemic myocardial blood flow in the stenotic territory; *Q_N max_*, hyperemic myocardial blood flow in the normal territory; *R_S max_*, hyperemic myocardial resistance in the stenotic territory; *R_N max_*, hyperemic myocardial resistance in the normal territory).

*FFR* is a reliable index for assessing the physiological significance of coronary artery stenosis [5]. However, *FFR* uses pressure instead of blood flow; therefore, achieving maximum hyperemia is a prerequisite for measuring *FFR* [6]. Hyperemia is defined as the condition of forward flow only and low hydrostatic pressures. Hence, the need for medication and pressure measurement wires arises in measuring *FFR*, leading to an extension of the procedure time and an increase in cost [2].

The Stewart–Hamilton equation was used to measure the blood flow velocity based on the rate of change in the substance concentration following the introduction of a known amount of the substance into the bloodstream, as illustrated in the formula below [7]. This method is also used to measure cardiac output [8,9].
$$Blood\;flow=\frac{I}{\int_{0}^{\infty }C\left(t\right)dt}$$
where *I* is the total amount of injected contrast agent, the integral of *C(t)dt* is the area under the intensity change curve of the contrast agent registered by coronary angiography, and *t* = 0 is the injection time.

CCIA was conducted using the Stewart–Hamilton equation rather than pressure, as indicated in the formula below.
$$CCIA=\frac{Q_{N\;ave}}{Q_{S\;ave}}=\frac{\frac{I}{\int C_{N}dt}}{\frac{I}{\int C_{S}dt}}=\frac{\int C_{S}dt}{\int C_{N}dt}=\frac{AUC_{S}}{AUC_{N}}\;$$

(*Q_S ave_*, average myocardial blood flow in the stenotic territory; *Q_N ave_*, average myocardial blood flow in the normal territory; *I*, amount of injected contrast agent; *C_N_*, change in the intensity of the contrast agent over time in the normal territory; *C_S_*, change in the intensity of the contrast agent over time in the stenotic territory; *AUC_S_*, area under the curve in the stenotic territory; *AUC_N_*, area under the curve in the normal territory).

In this study, we introduce a new approach called *CCIA*, which computes the blood flow velocity ratio by analyzing alterations in the contrast agent concentration over time during coronary angiography.

The physiological factor, *CCIA*, represents the ratio of the average blood flow in the normal arteries to that in stenotic arteries. The *CCIA* value is defined as the ratio of the area under the curve (*AUC*) of the normal arteries to that of the stenotic arteries. *CCIA* serves as an indicator of the physiological significance of coronary artery stenosis and directly measures coronary blood flow, in contrast to the *FFR*, which relies on pressure-derived values.

### 2.3. Schematic Diagram of the Stenosis Model

This study used nine ideal acrylic stenosis vessel models. These models featured varying stenosis severities of 30%, 50%, and 70%, with lengths of 6, 18, and 30 mm for the stenosis segments. Figure 1A illustrates the relevant geometric parameters. The detailed geometric dimensions are listed in Table 1.

### 2.4. The In Vitro Coronary Circulation System

An in vitro coronary circulation system (Figure 1B) was designed to replicate the physiological coronary circulation. A Windkessel model was used, which incorporated an air tank to regulate the blood flow, pressure waveforms, and phase differences to mimic those observed in the coronary arteries [10]. The setup consisted of a silicone tube with a diameter of 3 mm. The perfusate used in the experiment consisted of a mixture of Doppler fluid (Model 707, ATS Laboratories, Bridgeport, CT, USA) and glycerin with a viscosity of 4.5 cP. A pulsatile pump (Model 55-3305, Harvard Apparatus Corp., Holliston, MA, USA) was used to generate an average flow rate of 25 cm/s within the range of the physiological flow rate of the coronary arteries under non-stenotic conditions. The output phase ratio was set to 60% with a heart rate of 60 bpm to ensure a physiological pulsatile flow.

### 2.5. Pressure Data Acquisition

To evaluate the accuracy of the *CCIA*, we conducted *FFR* measurements within an in vitro coronary circulation system. For the experimental *FFR* assessments, we measured the pressure at both the proximal and distal sites. Pressure measurements were obtained using a 3 French pressure sensor-tip guidewire (Combo Wire XT; Volcano Corporation, San Diego, CA, USA) through a catheter. Measurements were taken 20 mm proximal to the stenosis site with the catheter inserted 500 mm proximal to the stenosis location. The *FFR* was computed as the average pressure over 10 cycles.

### 2.6. The Angiography Protocol

Coronary angiography images were captured at 30 frames per second using iodixanol (Visipaque) on an angiographic system (Artis Zee, software version VD11 C 180404, Siemens, Forchheim, Germany). To ensure adequate mixing of the contrast agent and fluid, we injected 4 mL at a rate of 16 mL/s using an automatic contrast agent injector 500 mm from the stenosis model. Subsequently, we obtained a 300-frame coronary angiography image over a 10 s duration with an automatic contrast agent injector. The coronary angiography images were collected in triplicate at specified time intervals, and data were acquired using the same methodology while varying the stenosis model (Figure 2).

### 2.7. Image Processing

Coronary angiography images were acquired using the same in vitro coronary circulation system, utilizing nine distinct stenosis models with varying degrees of stenosis and stenotic lengths. These images were processed offline using MATLAB (version 2021a, Mathworks, Natick, MA, USA) to calculate the *AUCs* for the *CCIA*. The stenotic area was defined as the region in which the blood vessel narrows before returning to its normal width in the angiography. The proximal area was defined as a region equal in size to the stenotic area, beginning 10 mm from the end of the stenotic area. Angiography images encompassing both the proximal and stenotic areas were captured for 10 s, starting before the injection of the contrast agent and lasting until the contrast agent dissipated. The change in the contrast intensity over time initially peaked after the injection of the contrast agent and then gradually decreased. The *CCIA* value was derived by calculating the ratio of the values of the *AUC* in each respective area, as shown in Figure 3.

### 2.8. In Vivo Coronary Circulation 

The study protocol was approved by the Institutional Review Board of Jeju National University Hospital (reference number 2020-03-034). Among 15 patients, 15 *FFR* and 11 instantaneous wave-free ratio (iFR) measurements were compared and analyzed with *CCIA*. The coronary arteries are dynamic structures, requiring specialized imaging processing for *CCIA*. Coronary angiography images were obtained at a rate of 30 frames per second. Electrocardiography was used to precisely synchronize each angiographic frame with the cardiac cycle, ensuring accurate timing information. Subsequently, we performed pattern matching to confirm the coordinates of the coronary artery movement over time. Therefore, the analysis could only be conducted in patients with a normal sinus rhythm. These coordinates were then applied inversely to obtain static coronary artery images against the background, and we obtained the *CCIA* values as if from in vitro signals.

### 2.9. Statistical Analysis

Statistical analyses were performed using MATLAB software (version 2021a, MathWorks, Natick, MA, USA). Linear regression analysis was used to investigate the relationship between CCIA and FFR and the iFR in vivo and between FFR and CCIA in vitro. To understand how stenosis severity and length influence the relationship between FFR and CCIA, we built separate linear regression models for different stenosis conditions, with severities of 30%, 50%, and 70% and lengths of 6, 18, and 30 mm. This approach aimed to identify potential interactions between these factors. For the in vivo analysis, separate regressions were conducted to evaluate the correlations between CCIA and both FFR (*n* = 15) and the iFR (*n* = 11). Correlation analysis between groups was performed using Pearson’s correlation analysis. The values of continuous variables are means and standard deviations (SDs), and categorical variables are expressed as frequencies and percentages. Comparison of continuous variables between groups was performed using the independent sample t-test, and the categorical variables were assessed with a chi-square test. For each statistic, the significance level was less than 0.05. *p*-values < 0.05 were considered statistically significant for the in vivo and in vitro data.

## 3. Results

### 3.1. In Vitro Study

FFR measurements were conducted using nine distinct stenosis phantoms with varying degrees of stenosis and lengths within an in vitro coronary circulation system. The comprehensive results (Figure 4) serve as a valuable reference for the assessment of CCIA.

The *FFR* values displayed consistent trends related to stenosis length and severity (Figure 4). For a stenosis length of 6 mm, we observed a decrease in the *FFR* as the degree of stenosis increased. With 30%, 50%, and 70% stenoses, the *FFR* values were 0.90, 0.86, and 0.85, respectively. Similarly, the following trends were observed for a stenosis length of 18 mm: with 30%, 50%, and 70% stenoses, the *FFR* values were 0.84, 0.64, and 0.61, respectively. For a stenosis length of 30 mm, the *FFR* values were 0.69, 0.37, and 0.33 with 30%, 50%, and 70% stenoses, respectively. These consistent trends emphasize the decrease in *FFR* values with increasing stenosis length and severity, highlighting the sensitivity of *FFR* to variations in the stenosis parameters (Figure 4).

Figure 5 outlines the average *CCIA* values based on the stenosis length and severity, along with the standard deviations, to provide a comprehensive view of the results. For a stenosis length of 6 mm, the *CCIA* values exhibited the following patterns: with 30%, 50%, and 70% stenoses, the *CCIA* values were 0.82 (±0.007, *n* = 3, *p* < 0.001), 0.68 (±0.007, *n* = 3, *p* < 0.001), and 0.61 (±0.004, *n* = 3, *p* < 0.001), respectively. For stenosis lengths of 18 mm, the *CCIA* values were 0.78 (±0.052, *n* = 3, *p* < 0.001), 0.69 (±0.025, *n* = 3, *p* < 0.001), and 0.44 (±0.016, *n* = 3, *p* < 0.001) with stenoses of 30%, 50%, and 70%, respectively. For stenosis lengths of 30 mm, the *CCIA* values were 0.80 (±0.018, *n* = 3, *p* < 0.001), 0.64 (±0.006, *n* = 3, *p* < 0.001), and 0.40 (±0.026, *n* = 2, *p* < 0.05) with stenoses of 30%, 50%, and 70%, respectively. As shown in Figure 5, the *CCIA* values showed a trend similar to the *FFR* values. However, although the *CCIA* was related to the stenosis degree, it was less related to length, except for with 70% stenosis (for 70% stenosis, 6 mm length: 0.61 ± 0.004; 18 mm: 0.44 ± 0.016; 30 mm: 0.40 ± 0.026).

Figure 6A illustrates the linear correlation between *FFR* and *CCIA*. As the degree of stenosis increased, both indices decreased proportionally (R = 0.9442, *p* < 0.01). This robust correlation underscores the potential clinical utility of *CCIA* as a less invasive method for evaluating coronary stenosis. The consistent trends observed in the *FFR* and *CCIA* measurements emphasize the reliability and diagnostic accuracy of the *CCIA* in assessing stenosis.

### 3.2. In Vivo Study

Analysis of clinical data from 15 patients revealed significant correlations between CCIA and both FFR (R = 0.5775, *p* < 0.05) and the iFR (*n* = 11, R = 0.7578, *p* < 0.01). Detailed results, including the vessel type and the CCIA, iFR, and FFR values, are presented in Table 2.

## 4. Discussion

Our study demonstrates that CCIA provides a feasible and accurate means of diagnosing stenosis in CAD. We found a strong correlation between CCIA and the FFR, indicating that CCIA provides a diagnostic accuracy similar to that of FFR. 

In current clinical practice, FFR is considered the gold standard for evaluating the functional significance of CAD. FFR has been extensively validated in numerous clinical trials and has demonstrated reliability and accuracy in assessing coronary lesions. Its utility is evident in various aspects, including its simplicity and ease of use, direct assessment of lesion significance, and strong correlation with myocardial ischemia, thus providing valuable clinical decision support [6,11,12].

Despite its clinical utility, FFR has some limitations. The requirement for wires and pharmacological agents during the procedure can pose potential concerns, leading to procedural complications and increased overall costs [2]. Another notable limitation of FFR is its reliance on pressure measurements rather than direct assessment of coronary blood flow [13]. Although FFR yields valuable information, it is not equivalent to the direct measurement of coronary blood flow, which is an essential determinant of myocardial ischemia [14]. To overcome these challenges, clinicians use the iFR as an alternative to FFR.

The iFR is measured during the wave-free period of the cardiac cycle when the resistance to blood flow is low. Although the functional significance of CAD can be assessed without vasodilators, the iFR still utilizes a wire and employs a pressure-based evaluation method similar to that of FFR [11]. Particularly, pressure-based assessments, though informative, may not fully represent the true hemodynamic conditions within coronary circulation [15].

Corrected thrombolysis in myocardial infarction (TIMI) frame count (CTFC) is another physiological method used to assess the coronary blood flow during coronary angiography [16]. However, it focuses solely on measuring the number of cine frames required for the contrast dye to reach the distal landmarks in the coronary arteries. While CTFC provides information on the coronary blood flow dynamics, it does not provide information about the underlying coronary anatomy, such as stenosis severity. As a result, CTFC may not fully represent the functional significance of coronary lesions [17].

Another previously explored method is the thermodilution coronary flow reserve (CFR), based on the Stewart–Hamilton equation [8,17], which was also employed in our research. The Stewart–Hamilton equation describes the relationship between blood flow and indicator dilution. This technique involves injecting cold saline into the coronary artery and measuring changes in blood temperature to calculate the flow rates. Although CFR uses the thermodilution method for functional assessment, it requires additional invasive measurements and is technically more demanding than the standard coronary artery pressure measurement, i.e., FFR [18].

Recently, virtual FFR was developed using the Navier–Stokes equations and the Poiseuille equation. These values were derived by reconstructing the coronary artery using 3D images and creating an artificial blood flow using these equations [18]. However, virtual FFR using these equations is not equivalent to FFR, as it cannot fully reflect the CMVC and resistance [19]. In this study, the pressure-derived FFR decreased with longer stenosis lengths, and the blood-flow-derived CCIA values showed no significant change with stenosis length (50% stenosis). According to Ohm’s law, the pressure changes depending on the resistances connected in series. Thus, the CCIA value derived from observing the blood flow can reflect the CMVC and resistance values [20].

To overcome the limitations of the existing methods, we introduced CCIA and validated it in comparison with the FFR in in vitro and in vivo coronary circulation systems. By utilizing the contrast media used as an indicator in routine coronary angiography, we derived physiological factors by applying the Stewart–Hamilton equation. This new approach allowed us to evaluate coronary blood flow directly, less invasively, and in real time during coronary angiography.

Although this study introduced CCIA as a new index, which is a less invasive diagnostic method that directly assesses blood flow for stenosis in CAD, it is essential to recognize several inherent limitations of our study. Controlled experiments were performed by maintaining constant pulsatile flow conditions in the in vitro coronary circulation system. However, the flow dynamics may vary significantly in clinical and in vivo conditions because of the inherent complexity of human physiology and individual patient factors, such as arrhythmias. Hence, further experiments under various conditions are required to validate the robustness of this method. The symmetrical stenosis models used in our study, although they offered a comprehensive range of degrees and lengths of stenosis, could not entirely replicate the intricacies of real-world coronary lesions. Patients with CAD frequently present with irregular, non-uniform stenotic features. These complexities may not have been entirely captured by the idealized models employed in our experiments. In the in vitro coronary circulation system used for the CCIA, the blood vessels are rigid and immobilized, warranting additional image processing of human coronary arteries subjected to dynamic movement. Therefore, image processing techniques are essential to compensate for factors such as breathing and heart rate variability. Moreover, the number of patients analyzed in the in vivo system was very small. However, our study primarily aimed to establish the feasibility of using CCIA under controlled conditions. Rigorous clinical validation is imperative to substantiate the reliability and effectiveness of this method in real-world clinical scenarios involving patients with CAD. Further research and thorough testing are warranted to bridge the gap between our in vitro findings and clinical practice. These limitations should be acknowledged and thoroughly addressed in subsequent studies and, most importantly, when considering the potential implementation of CCIA in clinical practice.

## 5. Conclusions

This study proposes a new, less invasive method for diagnosing stenosis in CAD. In vitro experiments on stenosis confirmed that the contrast intensity ratio between the proximal and stenotic sites could predict the severity of stenosis. Our findings demonstrate the feasibility and accuracy of the CCIA, which provides a diagnostic accuracy similar to that of the current gold standard, i.e., FFR. Nevertheless, future studies should focus on more extensive clinical trials to validate the effectiveness of CCIA in diverse patient populations. Investigating the potential integration of CCIA into existing diagnostic workflows and protocols in real-world clinical scenarios is essential. Comparative studies directly comparing CCIA with other emerging less invasive diagnostic modalities, such as vFFR, can provide insights into the relative advantages and limitations of these approaches. Furthermore, assessing the cost-effectiveness and resource implications of implementing CCIA in routine clinical practice would be valuable for healthcare decision-makers. Research into refining and optimizing the CCIA technique, potentially through advancements in imaging technology or algorithmic enhancements, could enhance its diagnostic precision. Lastly, investigating the potential role of CCIA in guiding therapeutic interventions and treatment strategies for patients with CAD would be a crucial direction for future exploration.

## Figures and Tables

**Figure 1 diagnostics-14-01429-f001:**
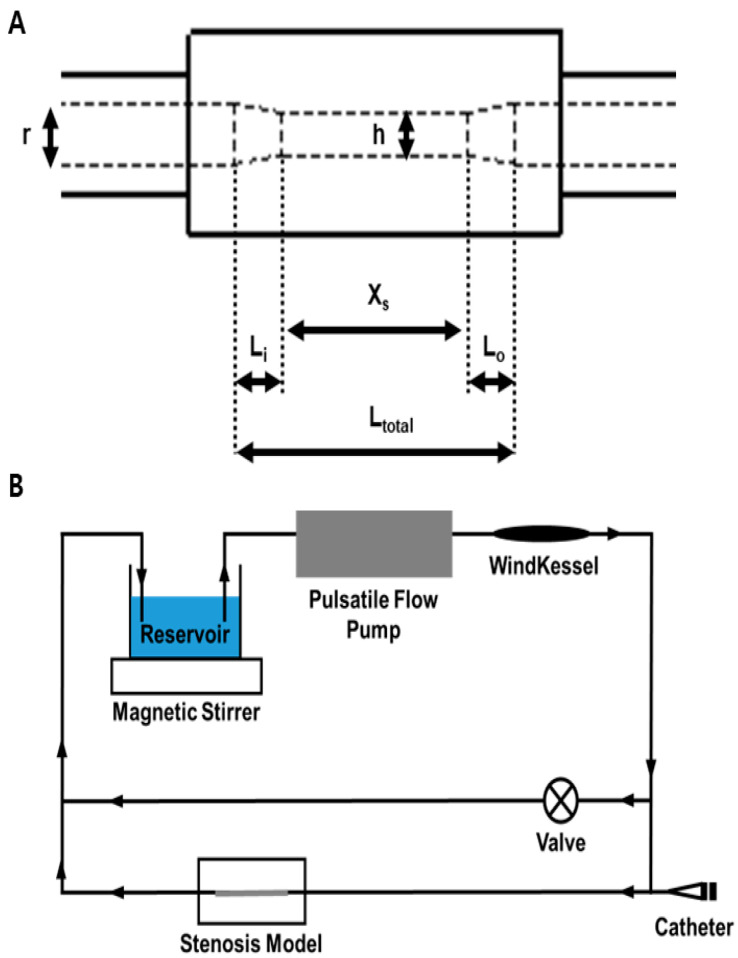
Schematic diagram of stenosis model and in vitro coronary circulation system. (**A**) Schematic diagram of stenosis model. h is the diameter of the stenotic vessel; L_i_ is the inlet length from the normal vessel to the front of the constant stenosis section; L_o_ is the outlet length from the constant stenosis section to the normal vessel; and L_total_ is the overall stenosis length; r is the diameter of the normal vessel (3 mm); X_s_ is the length of the constant stenosis section. (**B**) In vitro coronary circulation system. An in vitro coronary circulation system involves incorporating a Windkessel model that includes an air tank for controlling blood flow, mimicking pressure waveforms, and replicating phase differences similar to those observed in coronary arteries.

**Figure 2 diagnostics-14-01429-f002:**
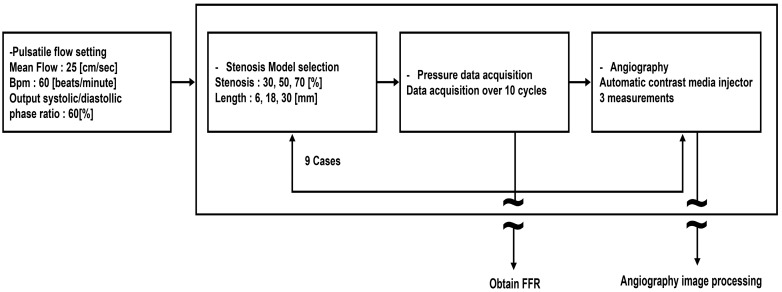
Experimental protocol. The stenosis model is changed while maintaining constant pulsatile flow. Pressure data are collected for FFR measurements, and coronary angiography is performed. BPM, beats per minute; FFR, fractional flow reserve.

**Figure 3 diagnostics-14-01429-f003:**
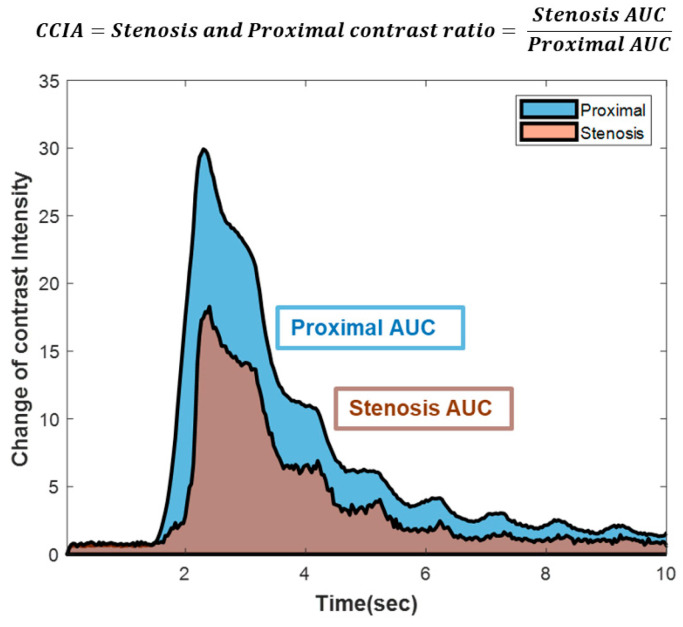
Example of AUCs in proximal and stenotic regions to conduct CCIA. This is an example of conducting the CCIA by assessing the change in the brightness of the contrast agent. The CCIA is computed as the ratio of the integral value representing the change in brightness of the contrast agent over time within the stenotic area to that within the proximal area. AUC, area under the curve; CCIA, coronary contrast intensity analysis.

**Figure 4 diagnostics-14-01429-f004:**
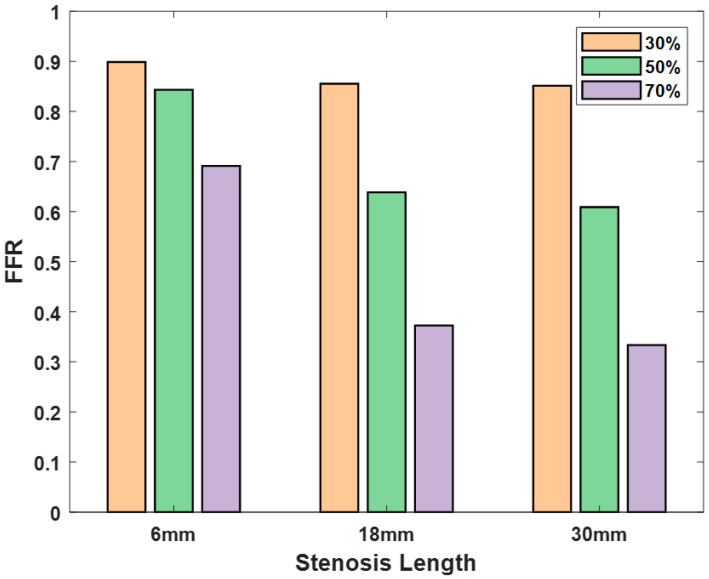
FFR measurements for nine stenosis models in the in vitro coronary circulation system. To evaluate the in vitro coronary circulation system, the FFR is measured by calculating the average pressure across 10 cardiac cycles. FFR, fractional flow reserve.

**Figure 5 diagnostics-14-01429-f005:**
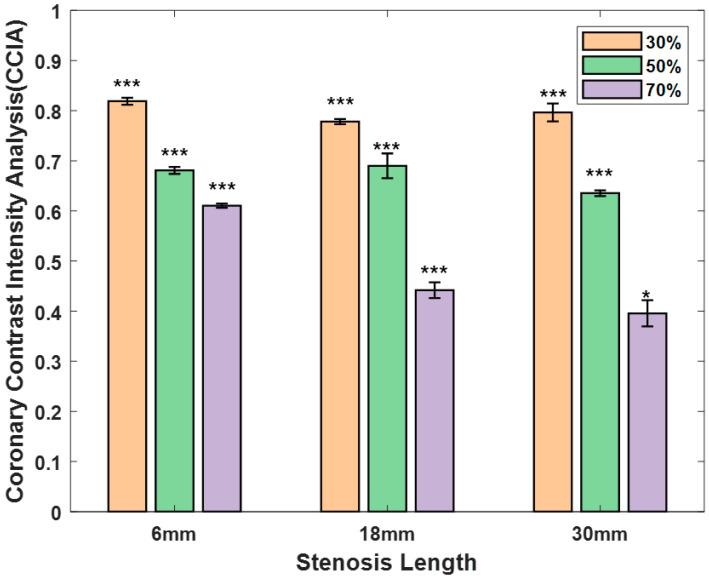
CCIA calculations for nine stenosis models in an in vitro coronary artery circulation system. (* *p* < 0.05. *** *p* < 0.001).

**Figure 6 diagnostics-14-01429-f006:**
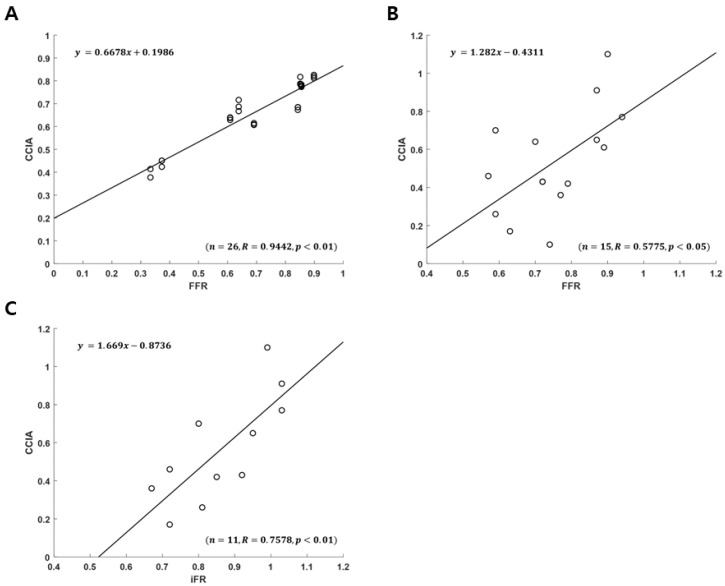
Correlation between FFR, iFR, and CCIA: (**A**) correlation between FFR and CCIA in vitro; (**B**) correlation between FFR and CCIA in vivo; (**C**) correlation between iFR and CCIA in vivo. The solid black line represents the line of best fit. CCIA, coronary contrast intensity analysis; FFR, fractional flow reserve; iFR, instantaneous wave-free ratio.

**Table 1 diagnostics-14-01429-t001:** Stenosis model geometric dimensions.

h/r	%DS	L_i_ (mm)	X_s_ (mm)	L_o_ (mm)	L_total_ (mm)
**0.3**	70	3	0	3	6
		3	12	3	18
		3	24	3	30
**0.5**	50	3	0	3	6
		3	12	3	18
		3	24	3	30
**0.7**	30	3	0	3	6
		3	12	3	18
		3	24	3	30

%DS, percentage diameter stenosis; h/r, nondimensional ratio; L_i_, inlet length from the normal vessel to the front of the constant stenosis section; L_o_, outlet length from the constant stenosis section to the normal vessel; L_total_, total length of the model (L_i_ + X_s_ + L_o_); X_s_, stenosis length.

**Table 2 diagnostics-14-01429-t002:** Summary of CCIA, iFR, and FFR values.

Patient No.	Vessel	Proximal Contrast Intensity	Stenosis Contrast Intensity	Distal Contrast Intensity	CCIA	iFR	FFR
**1**	LAD	1706	1041	1382	0.61	-	0.89
**2**	RCA	782	601	824	0.77	1.03	0.94
**3**	LAD	448	491	801	1.10	0.99	0.90
**4**	LAD	1007	657	288	0.65	0.95	0.87
**5**	LAD	1257	452	745	0.36	0.67	0.77
**6**	LCx	856	225	922	0.26	0.81	0.59
**7**	LCx	660	113	505	0.17	0.72	0.63
**8**	LAD	1282	824	1003	0.64	-	0.70
**9**	LAD	862	607	964	0.70	0.80	0.59
**10**	LAD	2371	1030	1258	0.43	0.92	0.72
**11**	RCA	354	162	287	0.46	0.72	0.57
**12**	LAD	519	220	431	0.42	0.85	0.79
**13**	LAD	212	91.89	104	0.43	-	0.72
**14**	LCx	477	48	794	0.10	-	0.74
**15**	LAD	416	379	470	0.91	1.03	0.87

CCIA, coronary contrast intensity analysis; FFR, fractional flow reserve; iFR, instantaneous wave-free ratio; LAD, left anterior descending artery; LCx, left circumflex artery.

## Data Availability

The data generated in this study can be made available by the corresponding author upon reasonable request.

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
