# Peer review of "A Novel Method for Angiographic Contrast-Based Diagnosis of Stenosis in Coronary Artery Disease: In Vivo and In Vitro Analyses"

_diagnostics, 2024, doi:10.3390/diagnostics14131429_

Round 1
Reviewer 1 Report
Comments and Suggestions for Authors
I read with interest the article by Woongbin Kang et al. titled: Novel Method for Noninvasive Diagnosis of Physiological Stenosis in Coronary Artery Disease: In Vivo and In Vitro Analyses.
I have some remarks to make.
The authors should be commended on providing insight into a novel technique to assess physiologic significance of a given stenosis, albeit still invasive and necessitating apparently high frame-rate acquisition which could expose the operators and patient to higher radiation. Also, although a novel technique in the armamentarium to assess stenoses in coronary arteries is always welcome, there are currently well-validated non-invasive techniques available on the market including CT-FFR. I still think there may be room for this technique in daily practice if extensively tested and validated.
One minor comment is the strong connotation of the acronym used in the text, CIA which reminds of central intelligence agency in the US. I suggest finding an alternative acronym.
It is interesting to note that there is strong correlation between the new technique and FFR in vitro studies but, although statistically significant, much weaker correlation in in vivo studies which apparently is where this technique will be tested. Given that fact that this technique was only tested in very specific length and diameter stenoses I have some reservations relying on this technique but I appreciate that it is a very early stage and any assumptions will be denied or accepted in future, high-volume studies.
In all, I believe that this is a well-written manuscript and provides very good amount of details and information on the derivation process. I expect to see this being tested on a bigger magnitude study but I believe there is room for this technique.
Comments on the Quality of English LanguageAcceptable quality, only minor editing.
Author Response
C1: “One minor comment is the strong connotation of the acronym used in the text, CIA which reminds of central intelligence agency in the US. I suggest finding an alternative acronym.”
R1: We agree with the reviewer’s comment. Thus, we have changed all CIA to CCIA.

Reviewer 2 Report
Comments and Suggestions for Authors
Manuscript describes the novel approach for evaluation of the coronary stenosis severity based on the results of coronary angiography and contrast intensity analysis. The results obtained warrant further clinical evaluation of the proposed method. And desire to be published. However, certain corrections need to be introduced in the present manuscript.
1. Authors refer to the proposed method as “noninvasive” both in the title and through the text. However, it is better to name it “less invasive” as they do in some parts, as it still requires injection of the contrast.
2. I recommend to evade the term “physiological stenosis”, as stenosis represents pathological condition.
3. In formula Line 93 – please, decipher dt
4. Line 97 – probably not, FFR, but CIA?
5. Line 218 – it is more correct to say: correlation coefficient (Spearman or Pearson???) was used to evaluate relationships between variables. “Simple correlation analysis” – this term is incorrect.
6. In statistics: were data checked for normality? How data are represented? Were means or medians calculated?
7. Figures 4 and 5 include both figure and table. Either table or figure should be presented. Number of repeats (n) should be indicated for each experiment. P-levels of differences should be indicated, at least where they were significant. Figure 4 – was the experiment performed only once? If so, it is not correct. If it was performed more than once, SD or interquartile range should be indicated.
8. Line 241 – Figure 5 not Figure 4???
9. Table 2 is not enough to evaluate clinical data. Medians (IQR) or means±SD for each method are required.
10. The results presented do not allow to evaluate method’s accuracy and reproducibility. Further research is required to state that. So, lines 283 – 284 in discussion should be changed.
Author Response
C1: “Authors refer to the proposed method as “noninvasive” both in the title and through the text. However, it is better to name it “less invasive” as they do in some parts, as it still requires injection of the contrast.”
R1: We agree with the reviewer. We have corrected our manuscript except for the tile.
Page 1 Title:
“Novel Method for Angiographic Contrast based Diagnosis of Stenosis in Coronary Artery Disease: In Vivo and In Vitro Analyses”
Page 8. Line 264
“This robust correlation underscores the potential clinical utility of CCIA as a less invasive method for evaluating coronary stenosis.”
Page 11. Line 336
“This new approach allowed us to evaluate coronary blood flow directly, less invasively, and in real-time during coronary angiography.”
Page 11. Line 371
“Comparative studies directly comparing CCIA with other emerging less invasive diagnostic modalities, such as vFFR, can provide insights into the relative advantages and limitations of these approaches.”
C2: “I recommend to evade the term “physiological stenosis”, as stenosis represents pathological condition”
R2: We agree with the reviewer. We have corrected our manuscript.
Page 1. Line 28
“CCIA is a promising alternative for diagnosing stenosis in patients with CAD.”
Page 2. Line 46
“To address these limitations, we propose an innovative and less invasive approach to diagnose stenosis”
Page 2. Line 50
“In this study, we aimed to assess the feasibility and accuracy of CCIA in diagnosing stenosis and compare CCIA with FFR.”
Page 2. Line 57
“We believe our study will introduce a more efficient and less-invasive diagnostic method using blood flow for stenosis in patients with coronary artery diseases and ultimately enhance patient comfort and reduce healthcare costs.”
Page 8. Line 266
“The consistent trends observed in the FFR and CCIA measurements emphasize the reliability and diagnostic accuracy of the CCIA in assessing stenosis.”
Page 10. Line 288
“Our study demonstrates that CCIA provides a feasible and accurate means of diagnosing stenosis in CAD. We found a strong correlation between CCIA and FFR, indicating that CCIA provides a diagnostic accuracy similar to that of FFR.”
Page 11. Line 339
“Although this study introduced CCIA as a new index, which is a less invasive diagnostic method that directly assesses blood flow for stenosis in CAD, it is essential to recognize several inherent limitations in our study.”
Page 11. Line 364
“This study proposes a new, less invasive method for diagnosing stenosis in CAD. In vitro experiments on stenosis confirmed that the contrast intensity ratio between the proximal and stenotic sites could predict the severity of stenosis.”
C3: “In formula Line 93 – please, decipher dt”
R3: We agree with the reviewer. We added content to the manuscript.:
Page 3. Line 94
“where I is the total amount of injected contrast agent, The integral of C(t)dt is the area under the intensity change curve of the contrast agent registered by coronary angiography, and t = 0 is the injection time.”
C4: “Line 97 – probably not, FFR, but CIA?”
R4: We agree with the reviewer. We have corrected our manuscript.
Page 3. Line 97
“FFR, which represents the average blood flow rate in coronary artery stenosis, CCIA was calculated using the Stewart-Hamilton equation rather than pressure, as indicated in the formula below.”
C5: “Line 218 – it is more correct to say: correlation coefficient (Spearman or Pearson???) was used to evaluate relationships between variables. “Simple correlation analysis” – this term is incorrect.”
R5: We agree with the reviewer. We have corrected our manuscript.
Page 6. Line 216
“Correlation analysis between groups was performed using a Pearson correlation analysis.”
C6: “In statistics: were data checked for normality? How data are represented? Were means or medians calculated?”
R6: We agree with the reviewer. We added content to the manuscript.
Page 6. Line 217
“The values of continuous variables are mean and standard deviation (SD), and categorical variables are expressed as frequency and percentage. The comparison of continuous variables between groups was performed using the independent sample t-test, and the categorical variables were assessed with a chi-square test. For each statistic, the significance level was less than 0.05.”
C7: “Figures 4 and 5 include both figure and table. Either table or figure should be presented. Number of repeats (n) should be indicated for each experiment. P-levels of differences should be indicated, at least where they were significant. Figure 4 – was the experiment performed only once? If so, it is not correct. If it was performed more than once, SD or interquartile range should be indicated.”
R7: We agree with the reviewer. In Figures 4 and 5, the tables were removed and only the figures were presented. The experiment in Figure 4 measured FFR and was performed only once. We measured FFR with pressure over 10 pulsatile pump cycles and is usually expressed as a single value by the clinically-used equipment (Combo Wire XT; Volcano Corporation, San Diego, CA, USA). Since this was from the in vitro mock circulation loop, the results were consistent which did not require statistical analysis.
Page 7. Line 230
Page 8. Line 258
Page 7. Line 244
“Figure 5. CCIA calculations for nine stenosis models in an in vitro coronary artery circulation system. (*p<0.05. ***p<0.001)”
Page 7. Line 246
“For a stenosis length of 6 mm, the CCIA values exhibited the following patterns: with 30%, 50%, and 70% stenoses, CCIA was 0.82 (±0.007, n=3, p<0.001), 0.68 (±0.007, n=3, p<0.001), and 0.61 (±0.004, n=3, p<0.001), respectively. For stenosis lengths of 18 mm, CCIA was 0.78 (±0.052, n=3, p<0.001), 0.69 (±0.025, n=3, p<0.001), and 0.44 (±0.016, n=3, p<0.001) with stenoses of 30%, 50%, and 70%, respectively. For stenosis lengths of 30 mm, CCIA was 0.80 (±0.018, n=3, p<0.001), 0.64 (±0.006, n=3, p<0.001), and 0.40 (±0.026, n=2, p<0.05) with stenoses of 30%, 50%, and 70%, respectively.”
C8: “Line 241 – Figure 5 not Figure 4???”
R8: We agree with the reviewer. We have corrected our manuscript.
Page 7. Line 245
“Figure 5 outlines the average CCIA values based on the stenosis length and severity, along with the standard deviations, to provide a comprehensive view of the results.”
C9: “Table 2 is not enough to evaluate clinical data. Medians (IQR) or means±SD for each method are required.”
R9: FFR and iFR values are expressed as a single value in the measuring instrument as the ratio of the average of several heart beat cycles.
C10: “The results presented do not allow to evaluate method’s accuracy and reproducibility. Further research is required to state that. So, lines 283 – 284 in discussion should be changed.”
R10: We agree with the reviewer. We have corrected our manuscript.
Page 10. Line 288
“CIA also showed high consistency and reproducibility, making it a reliable tool for diagnosing physiological stenosis.”
